# Ataluren for the Treatment of Usher Syndrome 2A Caused by Nonsense Mutations

**DOI:** 10.3390/ijms20246274

**Published:** 2019-12-12

**Authors:** Ananya Samanta, Katarina Stingl, Susanne Kohl, Jessica Ries, Joshua Linnert, Kerstin Nagel-Wolfrum

**Affiliations:** 1Institute of Developmental Biology and Neurobiology, Johannes Gutenberg-University of Mainz, 55122 Mainz, Germany; ananya.samanta85@gmail.com (A.S.); jlinnert@students.uni-mainz.de (J.L.); 2Institute of Molecular Physiology, Johannes Gutenberg-University of Mainz, 55122 Mainz, Germany; jries@uni-mainz.de; 3University Eye Hospital, Centre for Ophthalmology, University of Tuebingen, 72074 Tübingen, Germany; Katarina.Stingl@med.uni-tuebingen.de; 4Center for Rare Eye Diseases, University of Tuebingen, 72076 Tübingen, Germany; 5Institute for Ophthalmic Research, Centre for Ophthalmology, University of Tuebingen, 72076 Tübingen, Germany; Susanne.Kohl@med.uni-tuebingen.de

**Keywords:** Retinitis pigmentosa, Usher syndrome, patient-derived fibroblasts, translational read-through, TRID, Ataluren, ocular therapy

## Abstract

The identification of genetic defects that underlie inherited retinal diseases (IRDs) paves the way for the development of therapeutic strategies. Nonsense mutations caused approximately 12% of all IRD cases, resulting in a premature termination codon (PTC). Therefore, an approach that targets nonsense mutations could be a promising pharmacogenetic strategy for the treatment of IRDs. Small molecules (translational read-through inducing drugs; TRIDs) have the potential to mediate the read-through of nonsense mutations by inducing expression of the full-length protein. We provide novel data on the read-through efficacy of Ataluren on a nonsense mutation in the Usher syndrome gene *USH2A* that causes deaf-blindness in humans. We demonstrate Ataluren´s efficacy in both transiently *USH2A*^G3142*^-transfected HEK293T cells and patient-derived fibroblasts by restoring USH2A protein expression. Furthermore, we observed enhanced ciliogenesis in patient-derived fibroblasts after treatment with TRIDs, thereby restoring a phenotype that is similar to that found in healthy donors. In light of recent findings, we validated Ataluren´s efficacy to induce read-through on a nonsense mutation in *USH2A*-related IRD. In line with published data, our findings support the use of patient-derived fibroblasts as a platform for the validation of preclinical therapies. The excellent biocompatibility combined with sustained read-through efficacy makes Ataluren an ideal TRID for treating nonsense mutations based IRDs.

## 1. Introduction

Inherited retinal degenerations (IRDs) are rare genetic diseases of the retina that are characterized by vision loss or blindness. The most common among these rare diseases is retinitis pigmentosa (RP) with a prevalence of approximately 1:5000. RP represents a group of IRDs with the primary degeneration of rod photoreceptors, followed by the secondary loss of cone photoreceptors [1]. More than 50 genes have been associated with the non-syndromic forms of RP, in which only the retina is affected. In contrast, in syndromic forms, such as Usher syndrome (USH), Alström syndrome, Bardet–Biedl syndrome, and others, additional organs can be affected. RP belongs to major causes of visual impairment and blindness in young patients, although being a spectrum of orphan diseases. Over the years, several gene-specific approaches have emerged for overcoming the visual phenotype in IRDs [2,3,4]. The classical approach is gene augmentation that utilizes viral vectors, such as adeno-associated viral vectors (AAVs) or lentiviruses for the transfer of the wild type cDNA of the affected gene. Clinical trials using gene augmentation therapy led to promising results in a subgroup of Leber congenital amaurosis (LCA) that is caused by mutations in the *RPE65* gene [5]. In 2018, the AAV-based drug LUXTURNA^TM^ has been FDA-approved as a prescription gene therapy for patients with IRD and it is now also approved in Europe. However, the size of the coding sequences that exceed the cargo capacity of the currently applied viruses, e.g. IRDs caused by mutations in *USH2A*, hampered gene augmentation for several IRDs [2,6].

Mutations in *USH2A* are the most common cause in autosomal recessive RP [7,8], but they can also result in the human Usher syndrome (USH). USH is as a complex ciliopathy and the most common form of combined deaf-blindness [2,9,10]. Clinical USH is divided into three subtypes (USH1, USH2, USH3) based on the presence and progression of the clinical symptoms. USH1 is the most severe form and it is characterized by severe to profound congenital deafness, vestibular areflexia, and prepubertal onset of progressive RP. USH2 shows moderate to severe hearing loss, the absence of vestibular dysfunction, and later onset of retinal degeneration. USH3 is less common and displays progressive hearing loss, variable age of onset of RP, and variable vestibular impairment [9,11,12]. In most populations, one-third of USH patients present with the USH1 subtype, whereas two-thirds are classified as USH2. Among USH2 patients, mutations in the *USH2A* gene account for 55–90% of cases [9]. *USH2A* has a coding sequence of ~15.606 kb (GenBank NM_206933). Currently, the cargo capacity of clinically used AAVs is limited to 4.7 kb. Thus, USH2A´s coding sequence exceeds the cargo capacity of these AAVs by threefold. Therefore, an alternative therapeutic strategy for gene augmentation for USH2A patients is clearly needed.

Next-generation sequencing revealed that in-frame nonsense mutations cause between 5–70% of all genetic diseases [13]. In *USH2A*-related IRD, nonsense mutations account for ~16% of all *USH2A*-mutant alleles (199 nonsense mutations of 1234 USH2A-mutant variants, https://portal.biobase-international.com/hgmd/pro/gene.php?gene=ush2a). Nonsense mutations have an impact on protein translation [14] (Figure 1A). They turn a coding triplet into a stop codon, a so-called premature termination codon (PTC) that often results in a premature termination of translation (Figure 1B). In the case of a PTC, the resulting protein is shorter and often non-functional if translated at all. The transcripts are typically degraded by the mechanism of nonsense-mediated decay to avoid the accumulation of unfolded proteins, depending on the position of the nonsense mutation within a gene [13,14]. Translational read-through therapy is a valuable approach specifically targeting nonsense mutations [14]. This approach is based on the discovery that molecules, which are known as translational read-through-inducing drugs (TRIDs), bind to ribosomes, and thereby allow for the translation machinery to suppress a nonsense mutation. By this, a near-cognate tRNA binds to the A site of the ribosome, thereby inducing the elongation of the nascent peptide chain and, consequently, resulting in the synthesis of the full-length protein (Figure 1C). Yet, it has to be mentioned that a near-cognate tRNA mediates the elongation at the codon affected by the mutation and the amino acid at this position might differ from the amino acid in the wildtype protein [15,16] (Figure 1C; yellow circle). Thus, the functionality of the resulting protein has to be validated.

Especially in autosomal recessive disorders, even small amounts of functional protein are expected to be sufficient for therapeutic benefit, as heterozygous carriers of autosomal recessive alleles are healthy [2,17]. The ability of aminoglycoside antibiotics, such as Gentamicin, to induce read-through of nonsense mutations have been known for more than 50 years [14]. However, their long-term daily use is not practical due to severe side effects, namely retinal-, nephro-, and ototoxicity [18,19,20]. Due to these side effects, several attempts have been made to identify improved TRIDs. Out of these the frontrunner is Ataluren (PTC124, Translarna, 3-[5-(2-fluorophenyl)-[1,2,4]oxadiazol-3-yl]benzoic acid) [21]. To date, Ataluren under the commercial name Translarna^TM^ has been authorized for the treatment of Duchenne muscular dystrophy (DMD) and cystic fibrosis (CF) caused by in-frame nonsense mutations in the US, and has orphan drug status and conditional authorization for DMD in Europe [14]. Furthermore, Ataluren´s efficacy to induce read-through of in-frame nonsense mutations causing several ocular hereditary disorders, such as *USH1C*-related USH, has already been shown in several preclinical studies [14]. Last but not least, a phase II clinical trial for the treatment of aniridia, a genetic disorder that is often caused by nonsense mutations and is characterized by iris hypoplasia associated with additional ocular abnormalities is underway (NCT02647359).

In this study, we compared the capability of the aminoglycoside Gentamicin and Ataluren to induce translational read-through of the disease-causing nonsense mutation c.9424G>T;p.G3142* in the *USH2A* gene (GenBank NM_206933, Ensembl ENST00000307340.8), and demonstrate the read-through efficacy of Ataluren in transiently USH2A^G3142*^-transfected cell culture and in patient-derived fibroblasts.

## 2. Results

We analysed the relative abilities of Ataluren and Gentamicin to induce the translational read-through of a specific nonsense mutation (c.9424G>T; p.G3142*) in the *USH2A* gene. The *USH2A* gene is transcribed in at least two isoforms. The originally reported short isoform is an extracellular protein encoding of a 170 kDa USH2A protein [22]. Moreover, USH2A encodes for the ~580 kDa USH2A isoform b protein, being synonymously called Usherin [23]. The long USH2A isoform b is a transmembrane protein composed of a signal peptide, a large extracellular domain with several functional subdomains, such as FN3 (fibronectin type II motif) domains, a laminin G-like domain (LamGL), several laminin-type EGF (epidermal growth factor)-like modules (EGF-LAM) and two laminin G domains (LamG), a transmembrane domain, and the intracellular cytoplasmic tail domain containing a PDZ-binding motif (PBM) (Figure 2A). The PBM links the USH2A protein to several other proteins, such as whirlin (USH2D) and ninein-like protein (NINL) [24,25]. The USH2A protein is essential in the maintenance of photoreceptor cells and the normal development of cochlear hair cells [26] (Figure 2A). Numerous USH2-causing nonsense mutations in *USH2A* have been reported to date [27,28,29]. Specifically, HGMDpro (https://portal.biobase-international.com/hgmd/pro/gene.php?gene=ush2a) lists 199 nonsense mutations, which account for 16% of all *USH2A* mutations. Among them, the p.G3142* mutation was reported recurrently [27,30,31]. The p.G3142* mutation alters the triplet coding for a glycine (GGA) at codon 3142 into a PTC (TGA). On the protein level, the mutation is located in the extracellular FN3 18 domain.

### 2.1. TRIDs Induced Read-Through of the p.G3142* Nonsense Mutation in Transiently USH2A^G3142*^ Transfected Cells

We generated a USH2A reporter construct containing the extracellular FN3 domains 18–24 and 35 to analyse the read-through efficacy of Ataluren and Gentamicin on the p.G3142* mutation. Next, the p.G3142* pathogenic nonsense mutation was introduced into the USH2A deletion construct via site-directed mutagenesis. The mutated and wildtype *USH2A* cDNA were cloned into the pDisplay plasmid. The reporter constructs USH2A (3110-4052)_p.G3142* (USH2A^G3142*^, Figure 2B) and wildtype USH2A(3110-4052) (USH2A^+^) contain a HA-tag and a Myc-tag flanking the *USH2A* coding sequence 5´- and 3´-terminally, respectively, and a single transmembrane domain.

Subsequently, we evaluated the translational read-through of the USH2A p.G3142* nonsense mutation by Ataluren and Gentamicin in transiently USH2A^G3142*^-transfected HEK293T cells by indirect immunofluorescence staining and Western blot analyses. For this, the HEK293T cells were transfected with the reporter constructs and treated with Ataluren (10 µg/mL) or Gentamicin (1mg/mL) 6 h after transfection. The cells were processed 48 h later for further analysis. Indirect immunofluorescence staining was performed to demonstrate translational read-through of the TRIDs in a qualitative manner. For this, TRID-treated HEK293T cells that were transfected with the mutant reporter construct USH2A^G3142*^ showed an overlapping staining of anti-HA-labelling and anti-Myc labelling, which indicated a recovery of full-length protein expression of the mutant USH2A reporter construct (Figure 2C). In contrast, only a faint Myc-staining was detected in DMSO-treated USH2A^G3142*^-transfected cells, corresponding to low level of full-length protein expression. HA expression confirmed the transfection of the cells despite the weak Myc-staining (Figure 2C). In wildtype transfected cells (USH2A^+^), both antibodies showed a strong overlapping staining, indicating the correct expression of the full-length USH2A reporter protein.

Western Blot analyses were performed for quantitative analysis of the read-though efficacy. In Western blot analysis, the treatment of the USH2A^G3142*^-transfected cells showed a recovery of a 95-kDa band indicating a translational read-through of the nonsense mutation (Figure 2D). A faint 95-kDa polypeptide could occasionally also be observed in control (DMSO-treated) HEK293T cells transfected with the mutant construct. This product presumably represents spontaneous translational read-through of the nonsense mutation, as previously reported [18,32,33]. We normalized the intensity of the recovered USH2A protein band to the intensity of a loading control (actin) for the quantification of the read-through efficacy (Figure 2E). The quantification of protein expression revealed a significant increase in USH2A expression for both drugs as compared to DMSO-treated USH2A^G3142*^-transfected cells. The highest increase in the amount of full-length USH2A reporter construct was seen with Gentamicin (68.9-fold increase). Yet, Ataluren achieved an increase of 5.7-fold increase of USH2A protein expression in relation to the DMSO-treated cells (Figure 2E).

### 2.2. In silico Analysis of the Recovered USH2A Protein

The current hypothesis for the molecular mechanism of translational read-through is that the ribosomal A site accepts the binding of a near-cognate tRNA in the presence of TRIDs and subsequently incorporates an amino acid into the nascent polypeptide at the position of the PTC (Figure 1C) [13,16,34]. Consequently, the resulting restored proteins have insertion biases at the site of the PTC that might have an impact on the effectiveness of the protein and, subsequently, the therapeutic outcome. In the case of the present TGA nonsense mutation in *USH2A*, the amino acids arginine (R), leucine (L), serine (S), tryptophan (W), cysteine (C), and glycine (G) are predicted to be incorporated, with glycine being the wildtype amino acid residue (Table 1). Functionality studies would be helpful to prove whether the recovered USH2A protein is functional. However to date, no interaction partners of the extracellular FN3 domain of USH2A have been validated to proof the functionality of the recovered USH2A protein, such as previously performed GST-pulldowns to validate the functionality of the recovered harmonin/USH1C protein [18,20,35]. The possible effect of these introduced missense mutations on any resulting USH2A protein was analysed while using comprehensive *in silico* approaches, namely ALAMUT Genova, SIFT, and Polyphen-2 software, to address the limitation of this study. The *in silico* analyses predicted that the inclusion of any one of the possible amino acids would probably not be easily tolerated (Table 1). Yet, it is important to note that no missense (amino acid substitution) mutation at this amino acid residue has ever been observed in patients affected by USH or RP, and that only one of the potential amino acid substitutions, namely p.G3142R has ever been observed in population databases (i.e. gnomAD browser), and only once heterozygous (minor allele frequency 0.000004%). Consequently, this is not a polymorphic site.

We extended our research and analysed the recovered USH2A protein expression *in vitro* since our *in silico* analysis gave majoritarian damaging results. For this, we examined the subcellular localisation of the USH2A protein after TRIDs treatment and its capacity to rescue a cellular phenotype, namely primary ciliogenesis of primary cilia in patient-derived fibroblasts following TRIDs treatment.

### 2.3. Increased USH2A Protein Expression in Ataluren Treated USH2A_p.G3142* Patient-Derived Fibroblasts

The efficacy of translational read-through depends on the genetic background of the targeted mutation. Here, we investigated the translational read-through potential of TRIDs in fibroblasts that were obtained from an USH2A-RP patient. The patient is compound-heterozygous for the c.9424G>T;p.(G3142*) nonsense mutation and the in-frame deletion/insertion mutation c.13335_13347delinsCTTG;p.(E4445_S4449delinsDL). First, we performed Western blot analysis of fibroblasts from healthy donors, *USH2A* patient-derived cells, and TRIDs-treated *USH2A* patient-derived cells (Figure 3A). We detected a band at the expected size of ~ 350 kDa in healthy donor fibroblasts while using an antibody against the intracellular domain of the USH2A protein. Only a faint band corresponding to the USH2A protein was detected in untreated and Gentamicin-treated patient-derived fibroblasts. In contrast, Ataluren treatment restored USH2A protein expression in the USH2A-patient-derived cells (Figure 3A). We normalised the USH2A protein expression to the loading control actin (Figure 3B). The quantification of Western blot analysis revealed a significant increase in USH2A protein expression following the application of 5 µg/ml Ataluren (4.3-fold increase) as compared to DMSO-treated patient-derived cells. Only a weak increase of expression was detected following Gentamicin (1.3-fold increase) and 10 µg/ml Ataluren (1.9-fold increase) treatment. Normalisation of the relative USH2A expression levels in healthy control fibroblasts versus 5 µg/ml Ataluren-treated patient-derived cells revealed ~ 50% of restored USH2A protein expression in patient-derived fibroblasts (Appendix A).

We performed indirect immunofluorescence microscopy of healthy donor and USH2A patient fibroblasts to confirm the correct membrane localisation of USH2A. We detected high levels of membranous USH2A protein expression in healthy fibroblasts (Figure 3D). Untreated USH2A^G3142*^ patient-derived cells only showed reduced punctuated plasma membrane localisation of USH2A. The treatment of USH2A patient cells with Ataluren restored USH2A staining at the plasma membrane to control levels (Figure 3C). Although treatment with Gentamicin also resulted in an increased expression of USH2A when compared to control-treated patient-derived cells, the membrane localisation was weaker as compared to the Ataluren-treated cells (Figure 3C).

### 2.4. Primary Ciliogenesis in USH2A Patient-Derived Fibroblasts Treated with TRIDs

Recent studies suggested that Usher syndrome is a ciliopathy [2,38]. Thus, we analysed whether reduced USH2A protein expression alters ciliogenesis of primary cilia in USH2A patient-derived fibroblasts. We starved healthy control fibroblasts and USH2A patient-derived fibroblasts for 48 h to trigger ciliogenesis. Next, we determined the number of ciliated cells by immunofluorescence labelling while using the ciliary axoneme marker acetylated tubulin and the ciliary base marker pericentrin 2 (Figure 4A,B).

We observed that the number of ciliated USH2A patient cells was significantly lower than in the control healthy fibroblasts (Figure 4C; Appendix A). More specifically, 83% of healthy donor fibroblast cells (149 out of 198 cells) were ciliated, whereas only 54% of untreated patient cells (51 out of 97 cells) bear primary cilia. Interestingly, Ataluren treatment (5 µg/mL) was able to increase the number of ciliated cells to 78% in patient-derived cells (82 out of 119 cells). In contrast, no significant change in the number of ciliated cells was observed in Gentamicin (1mg/mL) or DMSO-treated patient-derived cells when compared to the untreated patient-derived cells. Importantly, treatment with Ataluren did not seem to have an effect on the number of ciliated cells in the control fibroblasts (Figure 4C).

## 3. Discussion

In this study, we show that Ataluren induces the translational read-through of the p.G3142* nonsense mutation in the human *USH2A* gene in HEK293T cells and in USH2A_p.G3142* patient-derived fibroblasts. USH2A is commonly mutated among the cohort of non-syndromic RP patients and USH2 patients. As mutations in *USH2A* account for 12%–25% of non-syndromic RP patients and for 55%–90% of USH2 patients, it is one of the most important genes in these rare diseases [9,39,40,41]. Moreover, 14%–29% of the disease-causing mutations in *USH2A* are nonsense mutations [42,43]; thus, targeting those mutations would be beneficial for a large cohort of affected individuals.

To date, there are only limited treatment options for IRDs, including patients with *USH2A* mutations. For selected genes, such as *RPE65*, gene augmentation using viral vectors is a therapeutic possibility, and gene augmentation using adeno-associated viruses (AAVs) represent the most promising approach for preventing photoreceptor cell degeneration in IRDs [44]. In 2017, the U.S. Food and Drug Administration approved the gene therapy while using AAVs for patients with confirmed biallelic *RPE65* mutation-associated IRD (LUXTURNA^TM^; Spark Therapeutics). Gene augmentation is not an option for several other more frequent forms of IRD (i.e. due to mutations in *ABCA4* or *USH2A*) [2]. One reason for this is that the coding sequences are extremely large, e.g. *ABCA4* or *USH2A* encodes for cDNAs of 6.8 kb and 15.606 kb, respectively, exceeding the packaging size (4.7 kb) of clinically-approved AAVs [2,45]. In addition, most of the genes are alternatively spliced (e.g. *USH1C, USH2A*) [12] and, to date, it is often not known which isoform must be replaced [46]. Thus, the investigated pharmacogenetic translational read-through treatment in the present study is a promising alternative to gene augmentation therapy at least for nonsense mutations.

In our study, the read-through efficacy of Gentamicin was much higher when compared to Ataluren in transiently USH2A^G3142*^-transfected HEK293T cells. In contrast, the read-through efficacy of Ataluren was significantly better in patient-derived fibroblasts, as detected by Western blotting. We achieved ~50% USH2A protein expression in Ataluren-treated cells when compared to fibroblasts of healthy donors (Appendix A). Moreover, in patient-derived fibroblasts, following the Ataluren treatment the restored USH2A protein was shown to localise correctly to the cell membrane and an increase in the number of ciliated cells to that of healthy cells was observed. Although this difference between the HEK293T cell line and patient-derived cells is difficult to explain, it is consistent with previously published data comparing the efficacy of Gentamicin with the aminoglycoside-derivate NB30 (Goldmann et al., 2010). In the latter study, the efficacy of Gentamicin was much higher in HEK293T cells when compared to NB30, whereas its efficacy was decreased as compared to NB30 in organotypic retina cultures. It might be that Gentamicin is more effective in HEK293T cells when compared to other cells, including human fibroblasts. In addition, several studies demonstrated the enormous side effects of Gentamicin, including oto-, nephro-, and retinal toxicity; therefore, it is evident that treatment with Gentamicin is not a therapeutic option [20,47]. In contrast, Ataluren has shown a very good safety profile in phase 1 clinical trials [21]. Thus, the improved efficacy of Ataluren as compared to Gentamicin might be due to its improved biocompatibility in animals and human [18,20,21,35,48]. In addition, it is known that translational read-though is dependent not only on the type of stop codon, but it is also highly influenced by the surrounding sequences. This fact might have an impact on the efficacy of read-through in HEK293T cells as compared to patient-derived cells bearing the patient’s genetic background, including intron-exon structures in mind, which might influence the read-through efficacy. Furthermore, the observed different efficacies might be related to the mechanism of translational read-through. TRIDs are thought to interfere with ribosomal fidelity, so that a near-cognate aminoacyl tRNA can bind at the site of PTC, resulting in the incorporation of another amino acid in its place and allowing for the translation machinery to continue (Figure 1C) [49]. The likelihood of the amino acid being integrated at the PTC site varies depending on the stop codon and on the TRIDs applied [15,16]. The alternative amino acid might have an impact on the stability, localisation, and/or the function of the resulting USH2A protein. Our detailed *in silico* analysis suggested that none of the inserted amino acid residues are likely to be easily tolerated. In terms of physicochemical difference, serine is considered to be the least deleterious, followed by arginine and leucine. Recent characterization of translational read-through products from HEK293T cells following the exposure to Ataluren revealed the predominant insertion of arginine (~ 69%), followed by tryptophan (~28%) and cysteine (~0.7%) at the PTC UGA, the PTC that is present in our analysed patient-derived cells [16], while glycine would be the wildtype. In contrast, the treatment of HEK293 cells with Gentamicin alters the frequency of amino acid insertion [16]. The performed analyses demonstrate that only ~48% of the integrated amino acids were arginine, whereas ~ 27% and ~ 24% were cysteine and tryptophan, respectively. Thus, after Gentamicin-treatment, more than half (~51%) of the recovered USH2A proteins might have cysteine and tryptophan residue, which could have a damaging effect on the USH2A protein function. In contrast, in Ataluren-treated cells arginine is the most commonly inserted amino acid; hence, the likelihood of getting a functional USH2A protein higher, at least based on the SIFT prediction. This might explain our observation that Ataluren-treatment resulted in a close to normal cilia phenotype, whereas Gentamicin treatment did not have this rising effect on the number of cilia in USH2A patient-derived cells.

In a previously published analysis of translational read-through of fibroblasts that were derived from a patient suffering for Niemann-Pick A/B, the SIFT prediction was also in agreement with the biochemical results obtained, whereas the Polyphen-2 gave the opposite results [50]. Although, *in silico* prediction programs are frequently used, they may be erroneous in determining the specific effect of a likely pathogenic variant, yet [51]. Therefore, they should be interpreted with caution and should not be taken as a definitive approach [50]. Indeed, whenever possible, one should use patient-derived cells, such as muscle cells, fibroblasts, induced-pluripotent stem cells (iPSCs)-organoids, or cell lines derived from iPSCs, for analysing translational read-through efficacy and functionality of the recovered protein [52,53]. These cells might also serve as an excellent screening model for identifying the best drug for each patient and mutation independently. Ataluren might be the better choice than Gentamicin for the treatment of IRDs, since a better efficacy and biocompatibility in photoreceptor cells is expected, based on the results of previous and present studies.

In line with other preclinical and clinical studies, our data demonstrate the efficacy of Ataluren as a pharmacogenetics drug for the treatment of various disorders that are caused by nonsense mutations including several IRDs. It might be debatable whether the observed improvement of USH2A-protein expression after Ataluren-treatment would be enough to stop the vision loss in USH2A-patients. However, we still observed a clear impact on the number of ciliated cells after the exposure of Ataluren, since the treated patient-derived cells expressed nearly the same number of ciliated cells when compared to control, despite the low level of rescued protein expression. Our data, in line with previously published data, therefore adds to the evidence that only a small increase in functional protein might be enough to alter the pathophysiological phenotype and halt or slow down the progression of a recessive disease [13,52,54]. This evidence is further supported by Phase 2 trials of Ataluren in patients with DMD that demonstrate a significant delay in disease progression in the patients [55,56]. The preclinical data on Ataluren treatment in DMD models are comparable to the results shown herein, which suggests that Ataluren is a valuable small compound for different IRDs caused by nonsense mutations. We hope to obtain further supporting data from the ongoing Ataluren Phase II clinical study of patients with Aniridia (NCT02647359).

There is a clear need to identify novel therapeutic strategies, since the sense of vision is highly important for humans and its loss markedly affects the quality of life. As nonsense mutations in genes that are responsible for photoreceptor cell and retinal pigment epithelial function are a frequent cause of IRD, the translational read-through therapy is a promising therapeutic option. In line with previous publications, our present study provides a proof of concept that patient-derived cells are an important tool to be used for the preclinical testing of small molecules. Furthermore, we demonstrate that Ataluren might be a good candidate drug for the treatment of individuals with nonsense mutations in USH genes as well as in genes that are associated with non-syndromic RP.

## 4. Materials and Methods

### 4.1. Translational Read-Through Drugs (TRIDs)

Ataluren (PTC124) from Absource Diagnostics GmbH, Munich, Germany was dissolved in dimethyl sulfoxide (DMSO; Sigma-Aldrich, Deisenhofen, Germany). Gentamicin was purchased from Sigma-Aldrich, Germany.

### 4.2. Antibodies and Dyes

Polyclonal anti-Ush2a (cytoplasmic tail) antibodies that were used for Western blot analysis (WB; 1:1000) and immunostainings (IF; 1:200) were previously validated [57]. Commercially available antibodies were used, as follows: monoclonal anti-HA (rat) from Roche Diagnostics (Mannheim, Germany; WB 1:500; IF 1:50), anti-dsRed2 (WB 1:2000) Chromotec (Martinsried, Germany), mouse monoclonal anti-Myc from Cell Signaling (Danvers, MD, USA), monoclonal anti-acetylated tubulin from Sigma (Deisenhofen, Germany; IF 1:4000), and goat polyclonal anti-Pericentrin 2 (1:200) Santa Cruz Biotechnology (Heidelberg, Germany). For Western blotting, the secondary antibodies goat-anti-rabbit, goat-anti-mouse, goat-anti-rat; conjugated to Alexa680; (WB 1: 20,000) obtained from Molecular Probes (Darmstadt, Germany) were used. For immunocytochemistry, goat-anti-rabbit, donkey-anti-goat, goat-anti-mouse antibodies conjugated to Alexa488 or Alexa647 (IF 1:400), Molecular Probes (Darmstadt, Germany), respectively, were used and nuclear DNA was stained with DAPI (4’, 6-diamidino-2-phenylindole) (1 mg/mL) (Sigma–Aldrich, Germany).

### 4.3. DNA Constructs

A human USH2A cDNA containing the extracellular part was used for the generation of the reporter construct (forward primer: 5’-tatttacccgggccaagtgatataccaacaccc-3’, reverse primer: 5’-tattatgtcgacggtgttttgacaaacatcctact-3’). The wildtype construct was inserted into the p-Display vector (Addgene, Germany) by using Xma I and Sal I as the restriction sites. The nonsense mutation c.9424G>T;p.G3142* for USH2A was generated by using Quick Change Lightning Site-Directed Mutagenesis Kit, according to manufacturer’s protocol (Agilent Technologies, Waldron, Germany). Primers were: forward 5’-gttttccataggagatcatatcaaagaatgatgccatttggcttc-3’ and reverse 5’-gaagccaaatggcatcattctttgatatgatctcctatggaaaac-3’. The primers were designed using Agilent´s homepage (https://www.agilent.com/store/primerDesignProgram.jsp).

### 4.4. Cell Culture

HEK293T (human embryonic kidney cells, expressing a mutant version of the SV40 large T antigene), were cultured in Dulbecco’s modified Eagle’s medium (DMEM) containing 10% heat-inactivated fetal calf serum (FCS) at 37 °C with 5% CO2. The cells were transfected with plasmids using Lipofectamine LTX and Plus Reagent (Invitrogen, Karlsruhe, Germany) according to manufacturer’s instructions. TRIDs were applied to the cells 6 h after transfection. Cells were harvested after 48 h of TRIDs treatment.

### 4.5. Fibroblast Culture

USH2A patient-derived fibroblasts with compound-heterozygous mutations in USH2A c.9424G>T;p.G3142* and c.13335_13347delinsCTTG;p.E4445_S4449delinsDL, as well as control healthy fibroblasts, were grown in DMEM-Glutamax (Invitrogen) containing 10% FCS and 1% penicillin-streptomycin at 37 °C with 5% CO2. TRIDs were added to the media the day after the splitting of the cells. The cells were harvested after 48 h of TRID treatment. For ciliogenesis, cells were grown in six-well plates with coverslips and serum-starved for 48 h with or without TRID treatment.

### 4.6. Immunocytochemistry

For the labelling of USH2A in the fibroblasts, the cells were washed with PBS (phosphate buffered saline) and permeabilized with 0.1% Triton X-100 for 10 min. [58]. After a washing step with PBS followed by 30 min. incubation with blocking solution [0.5% fish gelatine, 1% ovalbumin, in 1x PBS (27.4 mM NaCl, 0.6 mM KCl, 1.6 mM Na2HPo4, 0.4 mM KH2Po4 in ddH2O, pH adjusted to 7.4)], fibroblasts were incubated with anti-Ush2a (cytoplasmic tail) antibodies overnight at 4 °C. Following two washing steps with PBS, the fibroblasts were fixed with 2% paraformaldehyde (PFA) for 15 min. Following a washing step with PBS, the samples were incubated with fluorescent-labelled secondary antibodies and DAPI for 1 h at room temperature, washed with PBS, and mounted in Mowiol (Roth, Karlsruhe, Germany).

The HEK293T cells were washed in PBS and then fixed with ice-cold methanol for 5 min. Fibroblasts were fixed with 2% PFA for the staining of the cilia. The fixed cells were washed 3x with PBS and permeabilized with 0.1% TritonX-100 for 10 min., followed by 30 min. incubation in blocking solution. Primary antibodies were incubated overnight at 4 °C. On the next day, after washing with PBS, the samples were incubated with fluorescent-labelled secondary antibodies and DAPI for 1 h at room temperature. After washing, the coverslips were mounted in Mowiol (Roth, Karlsruhe, Germany).

### 4.7. Western Blot Analysis and Quantification

Fibroblasts were lysed in 0.5% Triton X buffer (50 mM Tris (pH 7.5), 5 mM EGTA, 150 mM NaCl, 1% Triton X-100, protease inhibitors) and the protein lysates were subjected to SDS PAGE gel electrophoresis, followed by Western blot. Western blots were analyzed while using the Odyssey infra-red imaging system (LI-COR Biosciences, Lincoln, NE, USA). For the quantification of the three experiments performed, the relative band intensities of the USH2A protein were normalized to the relative band intensities of the corresponding actin loading control.

### 4.8. In Silico Analysis

*In silico* predictions were performed to predict whether amino acid substitutions in the USH2A protein likely have an impact on the protein. SIFT (http://sift.jcvi.org/) prediction is based on the sequence homology of closely related sequences and the physical features if the integrated amino acid [37]. The score ranges from 0 (damaging) to 1 (tolerated). Polyphen-2 (http://genetics.bwh.harvard.edu/pph2/) is based on structural and comparative evolutionary considerations of the analysed protein. The score ranges from 0 (benign) to 1 (damaging) [50,59]. Additionally, the putative missense variants potentially incorporated at amino acid residue G3142 were evaluated by the integrated software Alamut Genova (http://www.interactive-biosoftware.com) while using default settings.

### 4.9. Microscopy and Image Processing

The immunofluorescence staining of specimens was analysed on a Leica DM6000B microscope using LAS-AF software (Leica, Bensheim, Germany). The following objectives were used: 506174: ∞/0.17/0, HCX PLANAPO,60×/0,75 PH2 and 506182: ∞/0.17/0, HCX PL APO, 63×/1,32O/L PH3CS. Fluorescence images processing was done with LAS-AF Leica imaging software and ImageJ/Fiji software [60,61].

### 4.10. Statistical Analysis

Statistical analyses were performed while using MS-Excel. The Student’s t-test was performed to prove the significance of observed differences (unpaired, two-tailed, assuming equal variance). A *p*-value of 0.05 and below was considered to be significant. ImageJ 1.52p software (Wayne Rasband National Institutes of Health, Bethesda, Maryland, USA; http://imagej.nih.gov/ij) was used for Western Blot and immunofluorescence quantification. Error bars are represented as standard deviation. At least three independent experiments were performed. The significance levels were set when *p* < 0.05 (∗), *p* < 0.01 (∗∗), *p* < 0.001 (∗∗∗).

## Figures and Tables

**Figure 1 ijms-20-06274-f001:**
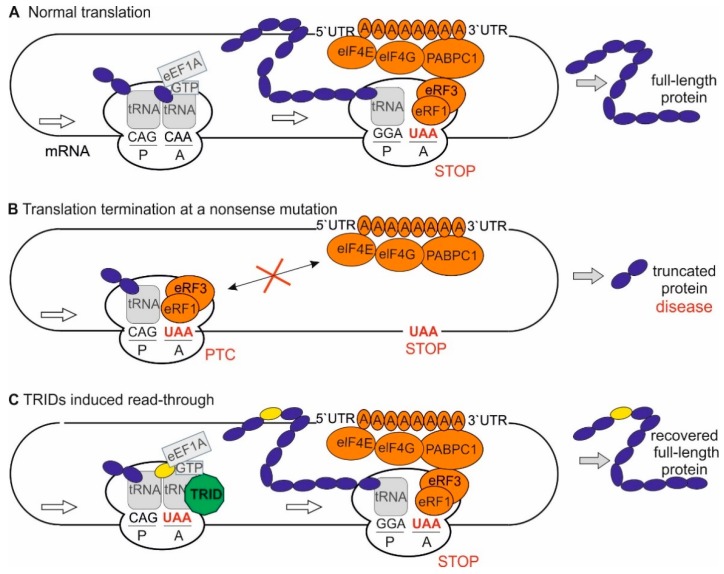
Translation: normal procedure, at a premature stop codon (PTC) and in the presence of translational read-through inducing drugs (TRIDs). (**A**) During elongation of the translation process, the binding of a cognate aminoacyl-tRNA (tRNA; grey rounded square) and the elongation factor eEF1A (grey rectangle) to the mRNA triplet at the A site catalyses the peptide bond formation between the nascent polypeptide (dark blue) at the P site and the new amino acid. At the site of a stop codon (red) a termination complex (eRF1, eRF3 GTP, eIF4E, eIF4G, PABPC1; orange) is formed. Upon formation of the termination complex, the release of the nascent polypeptide chain from the tRNA is induced. (**B**) In-frame nonsense mutations introduce a stop codon into the genomic sequence resulting in PTC in the mRNA. Translation of the mRNA stops (red X) resulting in a shortened polypeptide. This truncated polypeptide can have deleterious effects to the cells, including gain-of-function and loss-of function effects. (**C**) Translational read-through inducing drugs (TRIDs, green octagon) bind to ribosomes and can thereby enhance the translational read-through of PTCs. This results in the expression of full-length proteins. Resulting proteins might have an altered amino acid (yellow) at the site of the PTC.

**Figure 2 ijms-20-06274-f002:**
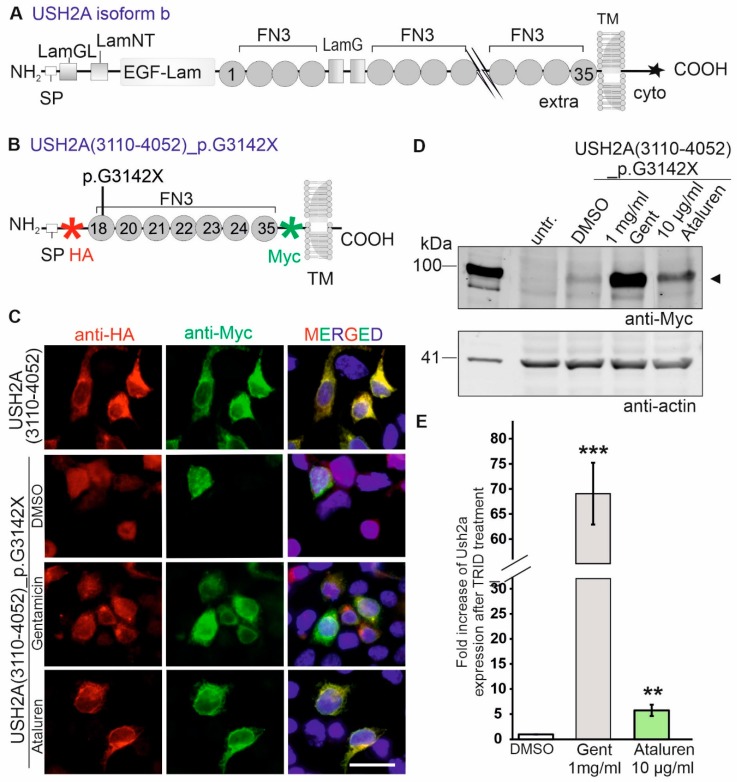
Ataluren induced translational read-through of the USH2A_p.G3142* nonsense mutation in transiently USH2A^G3142*^-transfected HEK293T cells. (**A**) Scheme of wildtype USH2A isoform b protein. Extra: extracellular domain; EGF-LAM: laminin-type EGF (epidermal growth factor)-like modules; FN3: fibronectin type II motif; intra: intracellular domain; LamG: laminin G domain; LamGL: laminin G-like domain; SP: signal peptide; TM: transmembrane domain; star indicates a PDZ-binding motif (PBM). (**B**) Scheme of reporter construct of USH2A carrying the p.G3142* nonsense mutation (USH2A31^G3142*^) used in present study. The reporter construct contains the extracellular FN3 domains 18-24 and 35. The coding sequence is flanked by an HA-tag and Myc-tag, respectively. (**C**,**D**) HEK293T cells were transiently transfected with the wildtype (USH2A^+^) and mutated USH2A (USH2A^G3142*^) reporter constructs. Six h later USH2A^G3142*^-transfected cells were treated with DMSO (control) Gentamicin (Gent, 1 mg/ml) or Ataluren (10 µg/µl). (**C**) Co-immunolabelling applying anti-HA (red) and anti-Myc antibodies (green) validated the translational read-through of the nonsense mutation after Gentamicin and Ataluren treatment in transiently transfected USH2A^G3142*^ HEK293T cells. Nuclei are stained with DAPI (blue). All images are in the same magnification, scale bar: 25 µm. (**D**) Western blot analysis with anti-Myc antibody detected full-length USH2A expression (∼98 kDa) after Gentamicin and Ataluren treatment of USH2A^G3142*^-transfected cells. Both TRIDs increases the expression of the mutated reporter construct (arrow head). Actin (42 kDa) was used as a loading control. (**E**) Quantification of the Western Blot analysis. Error bars represent the standard deviation, *p* < 0.001(***), *p* < 0.01 (**).

**Figure 3 ijms-20-06274-f003:**
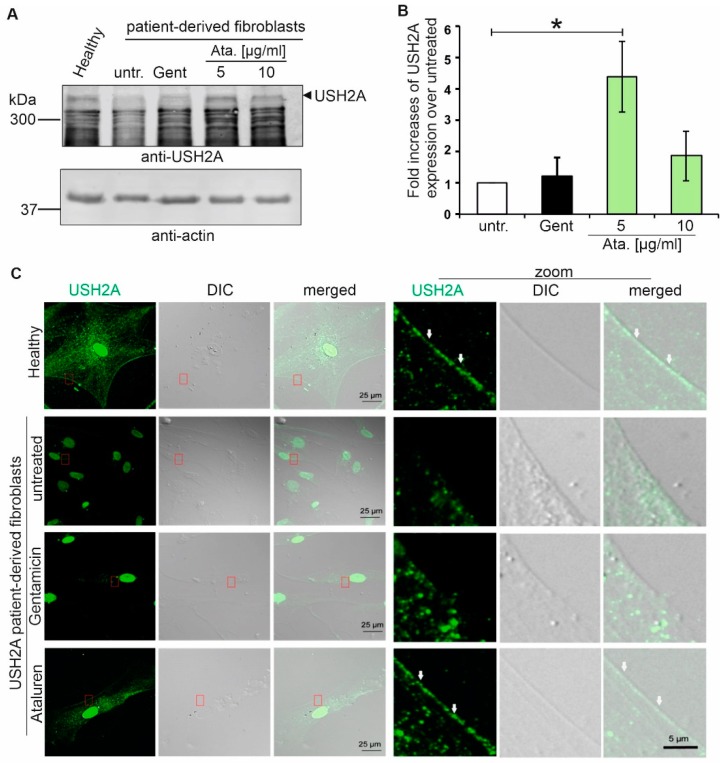
Ataluren induced translational read-through in patient-derived fibroblasts. (**A**) Fibroblasts of healthy donors, untreated USH2A patient-derived fibroblasts (untr.) and USH2A^G3142*^ patient-derived fibroblasts treated with Gentamicin (Gent, 1mg/ml) or Ataluren (Ata., 5 µg/µL, 10 µg/µL), respectively, were subjected to Western blot analysis. A stronger USH2A expression was detected in fibroblasts of healthy donors and patient-derived cells treated with 5 µg/µL Ataluren compared the untreated and Gentamycin-treated cells. (**B**) Quantification of Western blot analysis demonstrated the significantly increased USH2A protein levels in 5 µg/µL Ataluren treated USH2A patient-derived fibroblasts compared to untreated patient-derived cells (* *p* < 0.05). Three independent experiments are included. (**C**) Indirect immunofluorescence analysis of healthy fibroblasts and USH2A patient-derived fibroblasts. Immunofluorescence staining revealed a membranous USH2A localisation in fibroblasts of a healthy donor and in Ataluren-treated USH2A^G3142*^ patient-derived cells. A faint, punctuated USH2A protein expression was observed in untreated USH2A patient-derived cells. Treatment with Gentamicin partially restored USH2A localization at the membrane in patient-derived cells. Scale bar representative for the first three columns: 25 µm; scale bar representative for all zoom images: 5 µm. Three independent experiments were performed.

**Figure 4 ijms-20-06274-f004:**
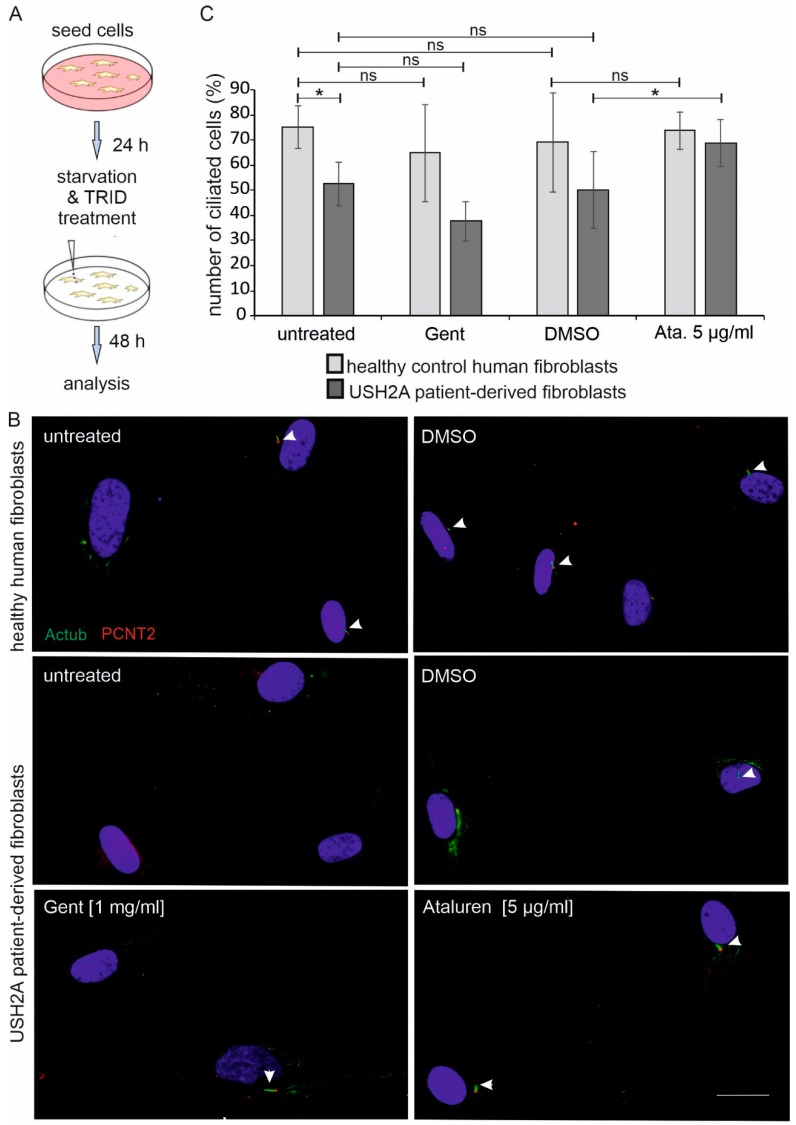
Ataluren increases the number of ciliated patient-derived fibroblasts. (**A**) Workflow. Healthy control human-derived fibroblasts and USH2A^G3142*^ patient-derived fibroblasts were seeded. 24 h later culture media was changed to starvation media. Cells were treated with dimethyl sulfoxide (DMSO), Gentamicin (1 mg/mL), or Ataluren (5 µg/µL), respectively. After 24 h cells were subjected for immunofluorescence analysis and the number of ciliated cells was counted. Four independent experiments were included. (**B**) Anti-acetylated tubulin (Actub, green) was used as a ciliary marker, Pericentrin 2 (PCTN2, red) as basal body marker and DAPI to stain the nucleus. All images are in the same magnification, scale bar represents: 25 µm. (**C**) The graph represents the percentage of ciliated cells versus the total number of cells in healthy control (light grey) and in patient-derived fibroblasts (dark grey), respectively. A significant decrease in the percentage of ciliated cells was observed in untreated, DMSO- or Gentamicin-(Gent)-treated USH2A^G3142*^ patient-derived cells versus healthy-donor derived fibroblasts. Ataluren treatment (5 µg/µl) restored the percentage of ciliated cells of patient-derived cells similar to that of healthy donors.

**Table 1 ijms-20-06274-t001:** *In silico* analysis of the functional effect of the predicted amino acid exchanges resulting from translational read-through of the p.G3142* nonsense mutation in *USH2A*. All variants were denoted based on the NCBI reference sequence for *USH2A* (NM_206933.2; GRCh38). Possible effects of the missense variants due to misincorporation during translational read-through protein function were assessed using the following computational *in silico* tools: MutationTaster (http://www.mutationtaster.org) [36], SIFT (http://sift.jvic.org) [37]; PolyPhen-2 (http://genetics.bwh.harvard.edu/pph2/), and the integrated software Alamut Genova (http://www.interactive-biosoftware.com) using default settings; gnomAD - Genome Aggregation Database (https://gnomad.broadinstitute.org/gene/ENSG00000042781 for USH2A.

					ALAMUT®					
Codon Position Altered	Possible Mispairing	Amino Acid Substitution	SIFT (0–1)	Polyphen-2 (0–1)	phyloP (-19.0;10.9)	Grantham Dist. (0–215)	Align GVGD [GV:353.86 - GD:0.00]	SIFT (Score: 0. Median: 3.71)	MutationTaster (probability 1)	gnomAD
UGA	GGA	p.Gly3142Gly (=wildtype)	-	-	-	-		-	-	0
	CGA AGA	p.Gly3142Arg	tolerated; 0.18	probably damaging; 1	2.3	moderate physiocochemical difference between G and R, 125	Class C0	deleterious	disease-causing, probability 0.999	1 heterozygous in 250,190 alleles
UGA	UUA	p.Gly3142Leu	damaging; 0.01	probably damaging; 1	5.63	moderate physiocochemical difference between G and L, 138	Class C0	deleterious	disease-causing, 0.999	0
	UCA	p.Gly3142Ser	tolerated; 0.09	probably damaging; 0.985	5.63	small physiocochemical difference between G and S, 56	Class C0	deleterious	disease-causing, 0.999	0
UGA	UGG	p.Gly3142Trp	damaging; 0	probably damaging; 1	0.13	large physiocochemical difference between G and C, 184	Class C0	deleterious	disease-causing, 0.999	0

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
