# Peer review of "Ataluren for the Treatment of Usher Syndrome 2A Caused by Nonsense Mutations"

_ijms, 2019, doi:10.3390/ijms20246274_

Round 1

Reviewer 1 Report

The authors used two experimental systems to examine the effects of gentamicin and Ataluren on the expression and subcellular distribution of USH2A protein, potentially as translational read-through therapy. In HEK2923T cells transfected with a truncated mutant USH2A reporter construct, both agents markedly elevated the reporter expression via apparently translational read-through. In patient-derived fibroblasts, Western blots and immunostaining suggested only Ataluren at 5 microgram/ml was effective. Overall, the design of the studies was innovative and the findings have potentially high clinical relevance. The rigor of the data, however, can be further improved. The authors are recommended to perform either or both of the two following experiments, so that the data will further strengthen the conclusion that Ataluren is a pharmacogenetics drug for the treatment of various disorders caused by nonsense mutation.

The anti-USH2A antibody used for Fig. 3 was not specific and not validated. Fig. 3B showed gentamicin had no effect on USH2A expression, yet the third row of Fig. 3C showed increased staining signal. Additional experiments will need to be performed with other anti-USH2A antibodies, hopefully validated antibodies. Studies in HEK293T cells used truncated USH2A. The effects of the two drugs on protein expression and distribution will be better tested after full length USH2A protein is expressed.

Reviewer 2 Report

This work was aimed at evaluating prospective effects of Ataluren in mediating the read-through on nonsense mutation in USH2A-related IRD. The experimental plan is minimal and concise, however the main results have the merit to identify Ataluren as putative pharmacological tools to ameliorate USH2A expression and membrane exposure when nonsense mutations are present.

The authors should only address minor revisions that may help to better interpret and justify their results.

1) Fig.2: HA-expressing cells are lower in USH2A-g3142 cells treated with DMSO. In other words, it seems that the efficiency of transfection is lower if compared to the other experimental conditions. The authors should address this issue, which may also be relevant in the evaluation of USH2A expression by Western blot in total lysates. The authors should also explain the reason why differences in USH2A expression between Gent and Ataluren observed in Western blot are not paralleled by alterations in immunofluorescence intensity.

2) The authors should explain the opposite behavior of Gentamicin and Ataluren in inducing USH2A protein expression in HEK293T-transfected cells and patient-derived fibroblasts. Why does Gentamicin appear to be more efficient in inducing USH2A in HEK293T, whereas its effect is only modest in fibroblasts? Similarly, why does ataluren slighty increase USH2A expression in HEK293T, whereas its effect is outstanding in patient-derived fibroblasts?

Round 2

Reviewer 1 Report

no further comments